# Production of *Trans*-Cinnamic and *p*-Coumaric Acids in Engineered *E. coli*

**Yuqi Liu [1], Weizhuo Xu [2,\*] and Wei Xu [2,\*]**

[1] School of Life Sciences and Biopharmaceuticals, Shenyang Pharmaceutical University, 103 Wenhua Road, Shenhe District, Shenyang 110016, China
[2] School of Functional Food and Wine, Shenyang Pharmaceutical University, 103 Wenhua Road, Shenhe District, Shenyang 110016, China
[\*] Correspondence: weizhuo.xu@syphu.edu.cn (W.X.); shxuwei8720@163.com (W.X.)

**Abstract:** *Trans*-cinnamic acid and *p*-coumaric acid are valuable intermediates in the synthesis of flavonoids and are widely employed in food, flavor and pharmaceutical industries. These products can be produced by the deamination of L-phenylalanine and L-tyrosine catalyzed by phenylalanine ammonia lyase or tyrosine ammonia lyase. Phenylalanine ammonia-lyase (PAL, EC 4.3.1.5) from *Rhodotorula glutinis* do not exhibit strong substrate specificity and can convert both L-phenylalanine and L-tyrosine. In this study, the PAL was utilized as the whole-cell biocatalyst, and the reaction conditions were optimized, and the production of *trans*-cinnamic acid and *p*-coumaric acid of 597 mg/L and 525 mg/L were achieved with high purity (>98%).

**Keywords:** *trans*-cinnamic acid; *p*-coumaric acid; phenylalanine ammonia-lyase; whole-cell biotransformation

## 1. Introduction

Phenylalanine ammonia-lyase (PAL, EC 4.3.1.5) is a key enzyme in the phenylpropane pathway of higher plants and functioned in the biosynthesis of various secondary metabolites, such as lignans, flavonoids and coumarins [1]. The enzyme is widely found in plants [2–7], fungi [8–12] and prokaryotes [13]. In plants, PAL is widely present in monocots [4], dicots [5], ferns [6] and algae [7]; in fungi, it mainly exits in *Saccharomyces* [9,10], *Ascomycetes* [11] and *Basidiomycetes* [12]; and in prokaryotes, such as *Streptomycetes* [13]. Some species of enzymes, such as *Rhodotorula glutinis*, can not only catalyze the non-oxidative deamination of L-phenylalanine (L-Phe) to *trans*-cinnamic acid and ammonia but also convert L-tyrosine (L-Tyr) to *p*-coumaric acid (Figure 1). Therefore, PAL has become an important therapeutic enzyme being used for the treatment of phenylalanine- and tyrosine-related complications in recent years [14].

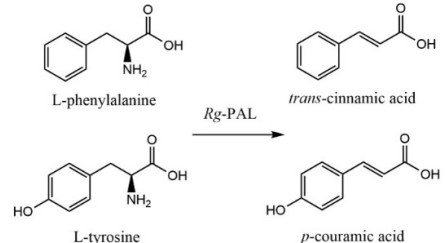

**Figure 1.** Reactions catalyzed by *Rg*-PAL.

PAL has a cofactor named 4-methylideneimidazole-5-one (MIO), which is a prosthetic group formed by the cyclization and elimination of the translated Ala-Ser-Gly tripeptide [14]. Two reaction mechanisms have been proposed for PAL to catalyze amino acid deamination. The first is the E1cb mechanism based on the electrophilic attack of the MIO moiety on the amino group of the substrate [15]. The second is the Friedel-Crafts's mechanism with electrophilic attack of MIO on the aromatic ring to eliminate $\alpha$-NH₃ and $\beta$-H from the substrate (Figure 2) [16].

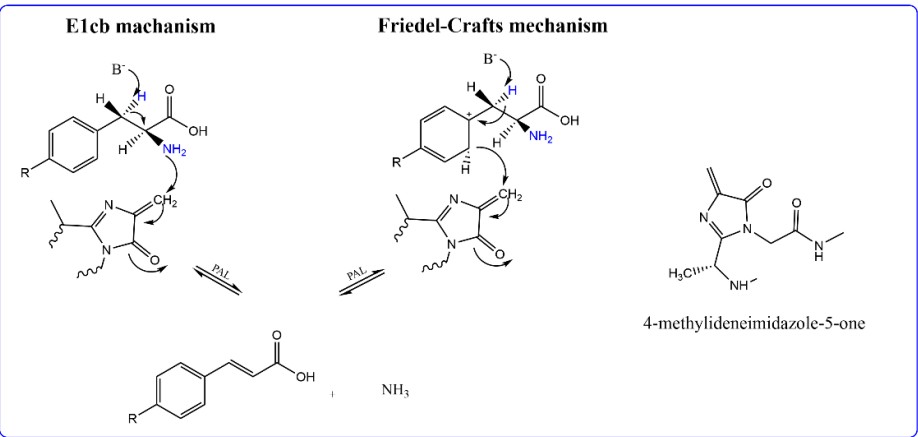

**Figure 2.** Reaction mechanism of PAL and the chemical structure of MIO cofactor.

*Trans*-cinnamic acid, the first product in the phenylpropane pathway, plays an important role in plant growth and development. *Trans*-cinnamic acid and its derivatives have been shown not only to be antioxidants but also to treat diseases, such as atherosclerosis and inflammatory damage [17]. Additionally, *trans*-cinnamic acid has been found to inhibit the formation of late glycosylation and products [18–20]. As an essential intermediate in the flavonoids synthesis pathway, *p*-coumaric acid exhibited antibacterial, anti-inflammatory effects and may contribute to the cardiovascular disease prevention [21,22]. Both *trans*-cinnamic acid and *p*-coumaric acid can be obtained via plant extraction, chemical synthesis and biosynthesis. The biosynthetic method has the advantages of low cost, high yield, sustainability and environmental friendliness. The promising microbial synthesis method was chosen to produce *trans*-cinnamic acid and *p*-coumaric acid in this work.

In previous work, Liang and co-workers also utilized *E. coli* BL21(DE3)/pMD18-*RgPAL* as a whole-cell catalyst to convert L-Phe and L-Tyr and investigated condition optimization obtaining 78.81 mg/L of *trans*-cinnamic acid and 34.67 mg/L of *p*-coumaric acid [23]. The recombinant *Zea mays* phenylalanine ammonia-lyase harboring *E. coli* BL21(DE3) was employed as whole-cell biocatalyst to transform L-Phe to *trans*-cinnamic acid in Zang' study, 5 g/L *trans*-cinnamic acid could be obtained from 10 g/L L-Phe under optimized conditions [24]. In Xue's experiment, Strain *DPD5124* expressing PAL/TAL from *Phanerochaete chrysosporium* and *DPD5154* expressing PAL/TAL from *Rhodotorula glutinis* were used as a whole-cell catalyst for the bioconversion of L-Tyr to *p*-coumaric acid, and the product yields of *p*-coumaric acid were achieved to 0.44 g/g dcw/L and 1.14 g/g dcw/L from 50 g/L L-tyrosine after optimizing reaction conditions [25]. The low solubility of tyrosine is an important reason for the low yield of *p*-coumaric acid. Based on our previous works, the catalytic activities of three phenylalanine ammonia-lyase from *Zea mays*, *Rhodotorula glutinis* and *Petroselinum crispum* were compared to transform L-Phe and L-Tyr. The results showed that the activity of converting L-Phe was *Pc*-PAL > *Rg*-PAL > *Zm*-PAL. Among these three enzymes, an excellent conversion of L-Tyr to *p*-coumaric acid was achieved by *Rg*-PAL, while the conversion of L-Tyr was extremely low or even absent with *Zm*-PAL and *Pc*-PAL as biocatalysts. Herein, Glycine-NaOH was chosen as reaction

medium and *Rg*-PAL harboring *E. coli* BL21(DE3) as whole-cell biocatalyst for the conversions of L-Phe and L-Tyr, while other reaction conditions were optimized.

## 2. Results

### 2.1. Optimization of Temperature

The reaction temperature is an important parameter affecting the catalytic reaction; either higher or lower temperature will affect the enzymatic efficiency, and appropriately increasing the temperature can reduce the viscosity of the mixture and enhance opportunities for the enzyme to collide with the substrate [26,27]. *Rg*-PAL was reacted with L-Phe and L-Tyr at different temperatures, ranging from 20 to 50 °C. The results are shown in the Figure 3. Although the enzyme was still highly active at 50 °C, the reaction temperature of 42 °C was finally chosen considering the stability and reusability of whole cells.

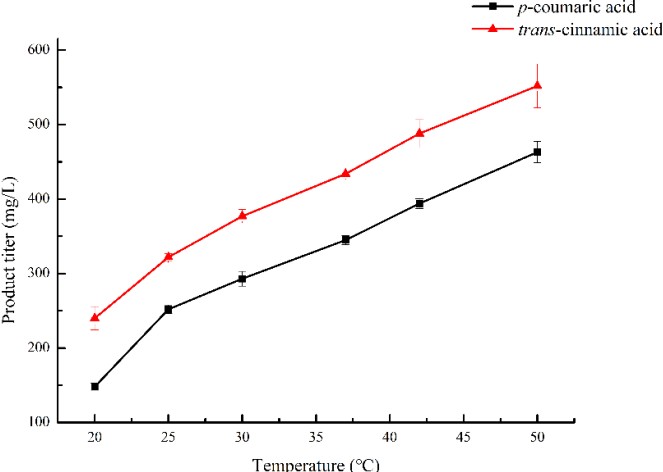

**Figure 3.** Effect of temperature on conversion of L-Phe and L-Tyr to *trans*-cinnamic acid and *p*-coumaric acid by *E. coli* BL21(DE3)/pETDuet-1-*RgPAL*. Reaction conditions: Time: 24 h; Medium: pH 9.0 10 mM Glycine-NaOH; L-Phe and L-Tyr 1 mg/mL; Cell amounts: 6.67 g/L.

### 2.2. Optimization of Time

The duration of whole-cell catalysis has an essential effect on the reaction. If the reaction time is too short, the reaction will be incomplete. With too long a reaction time, product inhibition may occur and affect products' yields. The experiment investigated the effect of conversion time on the synthesis of products and the results were shown in Figure 4. When L-Phe was the substrate, the peak yield of *trans*-cinnamic acid was 553 mg/L for 20 h; when L-Tyr was the substrate, the yield of *p*-coumaric acid was up to 393 mg/L for 24 h.

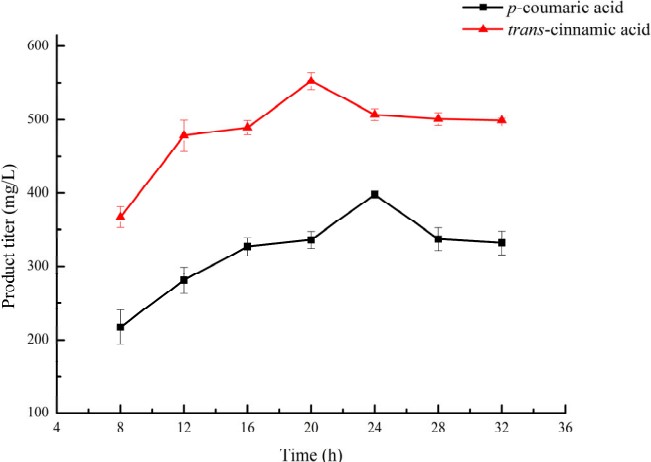

**Figure 4.** Effect of time on conversion of L-Phe and L-Tyr to *trans*-cinnamic acid and *p*-coumaric acid by *E. coli* BL21(DE3)/pETDuet-1-*RgPAL*. Reaction conditions: Temperature: 42 °C; Medium: pH 9.0 10 mM Glycine-NaOH; L-Phe and L-Tyr 1 mg/mL; Cell amounts: 6.67 g/L.

### 2.3. Optimization of Buffer Concentration

The amino acid deamination reaction occurs under alkaline conditions, and Glycine-NaOH was chosen as the reaction media in this study. Glycine, a non-polar amino acid, has both acidic and basic groups, is ionizable in water, and is highly hydrophilic. The polarity of the solution was changed by adjusting the addition of glycine in order to provide the ideal reaction environment for the substrate and enzyme. In this study, the impact of different glycine additions, ranging from 0 to 50 mM on the whole-cell catalytic reaction was evaluated at pH 9.0. Results were presented in Figure 5, the peak productions of *trans*-cinnamic acid and *p*-coumaric acid were 538 mg/L and 419 mg/L, respectively, at pH 9.0 20 mM Glycine-NaOH, respectively.

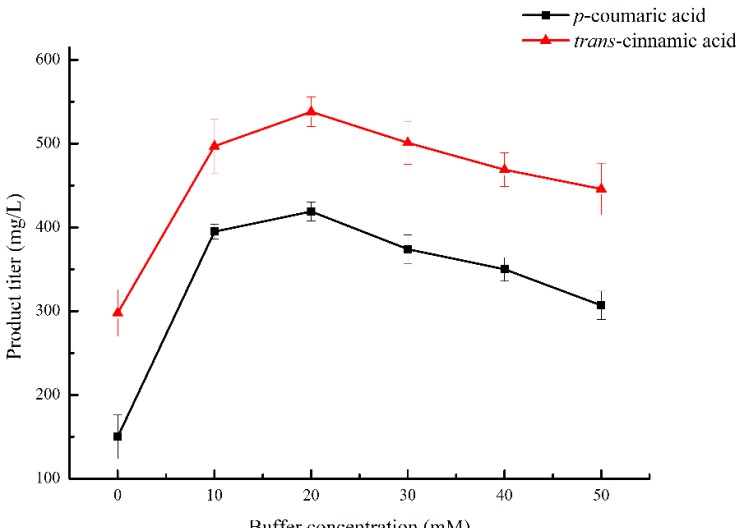

**Figure 5.** Effect of buffer concentration on conversion of L-Phe and L-Tyr to *trans*-cinnamic acid and *p*-coumaric acid by *E. coli* BL21(DE3)/pETDuet-1-*RgPAL*. Reaction conditions: Temperature: 42 °C; Medium: pH 9.0 Glycine-NaOH; L-Phe and L-Tyr 1 mg/mL; L-Phe conversion time: 24 h and L-Tyr conversion time: 20 h; Cell amounts: 6.67 g/L.

### 2.4. Optimization of pH

Changes in pH of the reaction medium can affect the charge density and molecular structure of the cell surface of enzyme molecules, resulting in changes in the rate and exit of substances into and out of the cell and in the catalytic efficiency of the enzyme. Therefore, it is important to regulate the pH of reaction medium. In this study, the effect of different pH (8.0–12.0) on the catalytic reaction of whole-cell was investigated. The results were shown in Figure 6. The maximum yield of *trans*-cinnamic acid was 552 mg/L at pH 10.0 and that of *p*-coumaric acid reached 455 mg/L at pH 11.0.

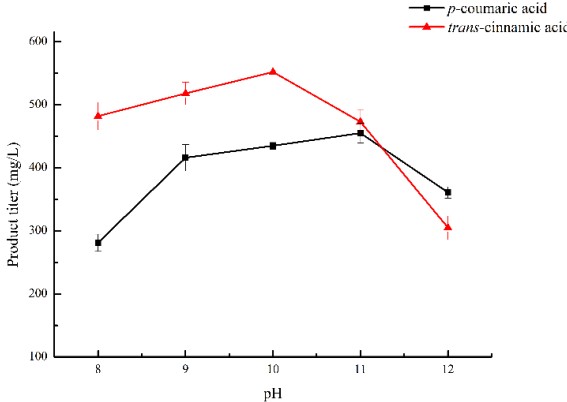

**Figure 6.** Effect of pH on conversion of L-Phe and L-Tyr to *trans*-cinnamic acid and *p*-coumaric acid by *E. coli* BL21(DE3)/pETDuet-1-*RgPAL*. Reaction conditions: Temperature: 42 °C; Medium: 20 mM Glycine-NaOH; L-Phe and L-Tyr 1 mg/mL; L-Phe conversion time: 24 h and L-Tyr conversion time: 20 h; Cell amounts: 6.67 g/L.

### 2.5. Optimization of Cell Amount

As a biocatalyst, the rate of reaction catalyzed by free cells directly depends on the concentration of cells, and, as shown from Figure 7, the rate of product generation catalyzed by cells increases linearly when the concentration of cells is relatively low. The production of *trans*-cinnamic acid and *p*-coumaric acid reached the maximum value of 595 mg/L and 525 mg/L when the amount of the cell increased to 10 g/L. The reaction rate decreased when the cell amount was higher than 10 g/L. The reason was that the cell concentration was too high and the cells clustered with each other, which reduced the opportunity for the active center of the enzyme in the cells to contact with the substrate, leading to a decrease in the catalytic reaction rate.

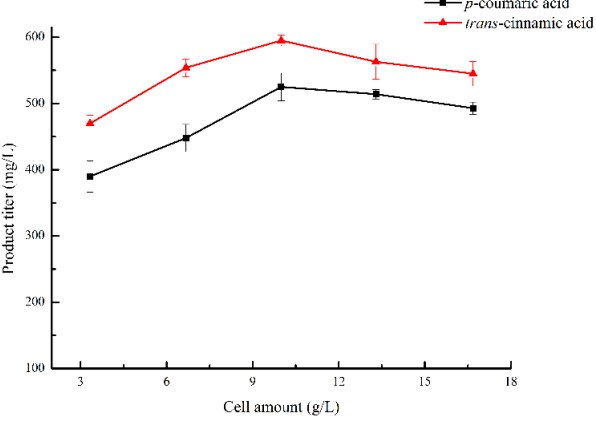

**Figure 7.** Effect of cell amounts on conversion of L-Phe and L-Tyr to *trans*-cinnamic acid and *p*-coumaric acid by *E. coli* BL21(DE3)/pETDuet-1-*RgPAL*. Reaction conditions: Temperature: 42 °C; L-

Phe conversion time: 24 h and L-Tyr conversion time: 20 h Medium: L-Phe in pH 10 20 mM Glycine-NaOH and L-Tyr in pH 11 20 mM Glycine-NaOH; L-Phe and L-Tyr 1 mg/mL.

### 2.6. Reusability

In the batch reactions, the cells were collected by centrifugation, washed with pH 7.4 10 mM PBS buffer three times and reused for the next batch after the previous reaction ended. As shown in Figure 8, *E. coli* BL21(DE3)/pETDuet-1-*RgPAL* can be used for at least five cycles. With the increasing use of whole cells, the yield of trans-cinnamic acid decreased from 597 mg/L to 294 mg/L and the production of *p*-courmic acid reduced from 525 mg/L to 172 mg/L. Therefore, *E. coli* BL21(DE3)/pETDuet-1-*RgPAL* is a promising strain for production of *trans*-cinnamic acid and *p*-coumaric acid.

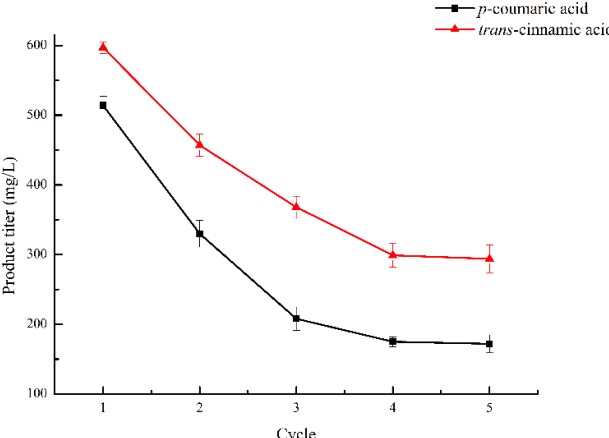

**Figure 8.** Batch reactions with repeatedly used *E. coli* BL21(DE3)/pETDuet-1-RgPAL cells at cell amounts of 10 g/L.

### 3. Discussion

Due to low solubility of L-tyrosine, Glycine-NaOH was chosen as the reaction medium to provide a more favorable reaction environment for the formation of *p*-coumaric acid by altering the glycine concentration. After optimizing reaction conditions, the product yields of *trans*-cinnamic acid and *p*-coumaric acid were 597 mg/L and 525 mg/L using free whole-cells as catalysts. Additionally, the catalytic activity of whole-cells remained well after five cycles under optimal reaction conditions. In further studies, the immobilized whole cells or enzymes as a catalyst will be utilized to produce *trans*-cinnamic acid and *p*-coumaric acid. Based on the optimized reaction conditions, we aim to further increase the enzyme activity through molecular modifications to achieve higher yields of products. Two reaction mechanisms of PAL have been reported so far. In the molecular docking results, some results were found to be more consistent with the Friedel–Crafts's mechanism, while others were more compliant with the E1cb mechanism (see in Supplementary Materials). According to this feature, targeted mutagenesis can be performed to improve the enzyme activity. In addition, the aim of improving their catalytic activity and reusability, as well as reducing the production cost, will be achieved.

## 4. Materials and Methods

### 4.1. Bacterial Strains and Plasmids

All bacterial strains and plasmids used in this study were listed in Table 1. *E. coli* DH5$\alpha$ was used to propagate plasmid. *E. coli* BL21(DE3) was used for functional expression of constructed plasmid. The *E. coli* BL21(DE3) harboring pETDuet-1-*RgPAL* was constructed by our laboratory.

**Table 1.** Strains and plasmids used in this study.

| Names | Characteristics | Source |
|:---:|:---:|:---:|
| Plasmid | | |
| pETDuet-1 | pBR322ori with P$_{T7}$; Amp$^R$ | Novagen |
| pETDuet-1-*Rg*-PAL | pETDuet-1 with *Rg*-PAL | This study |
| Strain | | |
| *E. coli* DH5$\alpha$ | $\Delta$Lac U169 ($\Phi$80 Lac Z $\Delta$M15) | Invitrogen |
| *E. coli* BL21(DE3) | F-ompT hsdS (rB-mB-) gal dcm (DE3) | Invitrogen |

### 4.2. Media and Cultivations

The engineered strains were precultured overnight at 37 °C in 3 mL LB liquid medium with ampicillin (100 mg/L), and then the preculture was inoculated into 30 mL LB liquid medium supplemented with ampicillin (100 mg/L) in 150 mL shake flasks at 1% of inoculum and grown at 37 °C again shaking at 180 rpm. When the culture reached an optical density of 0.6–0.8 at 600 nm, IPTG was added at a final concentration of 0.2 mM to induce gene expression and the cultivation was continued for additional 10 h at 37 °C and 180 rpm.

### 4.3. Whole-Cell Biotransformation of L-Phe to Trans-Cinnamic Acid and L-Tyr to p-Coumaric Acid by E. coli BL21(DE3)/pETDuet-1-RgPAL

*E. coli* BL21(DE3)/pETDuet-1-RgPAL was grown and induced with a final concentration of 0.2 mM IPTG. After induction at 37 °C for 10 h, the cells were collected by centrifugation and washed with 10 mM PBS buffer (8.0 g NaCl, 0.2 g KCl, 1.44 g Na$_2$HPO$_4$ and 0.24 g KH$_2$PO$_4$ dissolved in 1 L distilled water, pH 7.4) three times. Whole-cell biotransformation was performed in 150 mL Erlenmeyer flasks with 30 mL of the reaction mixtures containing Glycine-NaOH buffer, recovered cells and 1 mg/mL substrates under different conditions. In order to obtain higher yields of *trans*-cinnamic acid and *p*-coumaric acid, the reaction conditions were optimized. All experiments were carried out in triplicate and the mean values were calculated. The standard deviation for each test was calculated with excel and indicated as error bars.

### 4.4. Analytical Methods

Cell densities of cultures were determined by measuring their absorbance at 600 nm with a 2800 UV/visible spectrophotometer [UNICO (Shanghai) INSTRUMENT]. For analysis of *trans*-cinnamic acid and *p*-coumaric acid, the reaction mixtures were centrifuged at 3500× *g* for 10 min, and supernatants were collected to analyze the product concentration by HPLC using a Wondasil C18 Superb column (250 mm × 4.6 mm, 5 μm) with the column temperature of 30 °C. To detect the formation of *trans*-cinnamic acid, we used a solution (50% of 1% acetic acid and 50% of acetonitrile) as mobile phase at flow rate of 0.7 mL/min. The detection wavelength was set at 290 nm. To analyze the production of *p*-coumaric acid, the HPLC was eluted with the solution (60% of 0.1% formic acid and 40% of methyl alcohol) as mobile phase at flow rate of 0.6 mL/min. The detection wavelength was set at 310 nm. Meanwhile, the supernatants after centrifugation were adjusted to pH 4.2 by HCl and allowed to stand at room temperature for 30 min, then the acidified supernatants were centrifuged again to separate the *trans*-cinnamic acid precipitate. The supernatants after

centrifugation were adjusted to pH 4.6 by $H_3PO_4$ to obtain *p*-coumaric acid and other steps were same as the isolation of *trans*-cinnamic acid.

## 5. Conclusions

*Rg*-PAL-harboring *E. coli* BL21(DE3) as the whole-cell biocatalyst, the productions of *trans*-cinnamic acid and *p*-coumaric acid reached to 597 mg/L and 525 mg/L, respectively, via optimizing reaction conditions. After the cells were repeatedly tested five times, the enzyme still had good catalytic activity. This engineered strain is a potential strain to produce *trans*-cinnamic acid and *p*-coumaric acid for industrial applications.

**Supplementary Materials:** The following are available online at www.mdpi.com/article/10.3390/catal12101144/s1, Figure S1: The molecular docking results of PAL and L-Phe, Figure S2: Solubilities of L-Phe and L-Tyr in NaOH, Gly-NaOH and $Na_2CO_3$·$NaHCO_3$ at pH 10, Figure S3: Results of two rounds transformation of trans-cinnamic acids and p-coumaric acid, Figure S4: The standard curves for quantification of the product concentration, Figure S5: The HPLC graph of trans-cinnamic acid and p-coumaric acid.

**Author Contributions:** Y.L. (Data Curation, Investigation, Methodology, Writing—Original Draft); W.X. (Weizhuo Xu) and W.X. (Wei Xu) (Resources, Supervision, Writing—Review and Editing). All authors have read and agreed to the published version of the manuscript.

**Funding:** This research received no external funding.

**Data Availability Statement:** Data are available upon reasonable request.

**Conflicts of Interest:** The authors declare no conflict of interest.

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
