# Peer review of "Production of Trans-Cinnamic and p-Coumaric Acids in Engineered E. coli"

_catalysts, doi:10.3390/catal12101144_

Round 1

Reviewer 1 Report

The manuscript describes the production of p-coumaric acid and cinnamic acids starting respectively from tyrosine and phenylalanine employing engineered E. coli whole cells expressing Rg-PAL.

This work is not exceedingly original, since this reaction has been exploited (using Rg-PAL) since 1981 (Yamada, Shigeki, et al. "Production of L-phenylalanine from trans-cinnamic acid with Rhodotorula glutinis containing L-phenylalanine ammonia-lyase activity." Applied and environmental microbiology 42.5 (1981): 773-778.)

Thus,  a detailed and careful description and conduction of all the proposed experiments are necessary in order to highlight the novelty of the work.

Unfortunately, the manuscript is not very well organized and it is not easy to follow the reasoning behind the various statements. Some experimental procedures have been (wrongly) placed in the Result section (which is, by the way, lacking the title), such as par. 2.1 ("whole-cell biotransformation of L-Phe..."). The Discussion section contains statements that should be moved in the Introduction, such as the studies on the other PALs.

Moreover, here is a list of other issues to be solved:

- in the Abstract the productivity of the two acids is given as 597 and 525 mg/mL, whereas in the text is 597 (and 525) mg/L.

- page 1, row 33: "affinity group" is not a correct expression. Would it "prosthetic group" be OK?

- p2 r50: "experiment" --> "work"

- the choice of the buffer system (by the way, consider the substitution of "solvent" with "buffer" anywhere it makes sense) is not very clear. It is just a matter of pH or there are additional effects? Which ones? What is the solubility of L-Phe or L-Tyr at pH 10 in the proposed system? And in another buffer (e.g. carbonate, borate...) at the same pH?

- p3 r90: the authors state that the stability and reusability of the cells is optimized at 42°C, but they show no data about this parameter. Could they provide them, maybe in the Supporting Information if they think it would be too long?

- p5 r140: 9.99 g/mL couldn't be approximated to 10?

- I assume the product titer was calculated by HPLC employing a calibration line. Is it correct?

- Would it be possible to provide an HPLC trace of the end of the reaction? Again, in the SI file.

- Most important: since this procedure is proposed for preparation purposes, it would be welcome a product isolation procedure together with the characterization of the product and the calculation of an isolated yield.

Author Response

Dear Reviewer,

Thank you for your comments. Please find our revision in the attached files.

Reviewer 2 Report

Dear Authors

I agree an interesting optimization approach; very straight and clear; however, you state in the introduction that two mechanisms are discussed - later nothing in this direction is done. Overall, I feel you should show applicability by employing immobilized biomass to get proper reads/citations ... so I think it is a bit weak.

minor points:

- check italics style of all latin phrases and "E. coli".

- L44: correct "P-coumaric" to "p-Coumaric"

- Fig 1 has a poor resolution; same is true for Fig 3

- the cell amount and its possible effect is not clear; one need to show lower concentration; and if really cell aggregation or related is an issue one could mix just cells (not carrying the enzyme) with proper biocatalyst and see if this effect is observeable again!

- reusability is possible for 5 cycles but with huge loss of initial activity; this cannot be written as such a strong argument; why not putting cells in alginate to maintain productivity?

- you should isolate the product and give a proper yield. also as kind of conversion rate!

Author Response

Dear Reviewer,

Thanks for your instructive comments. Please find our revisions in the attached files.

Round 2

Reviewer 1 Report

The authors were able to correct a series of issues by accepting the suggestions from the Referees.

However, the manuscript should be further improved before publication:

- sentences in rows 45-46 ("which has... inflammatory") and r 48-49 ("The method... etc.") are problematic in terms of English language and should be reformulated;

- all along the manuscripts, substitute "reaction solvent" with "reaction medium";

- in Figure 2, in the Friedel-Crafts mechanism, please show the hydrogen attached to the benzene ring in position 2, where the curved arrow pointing towards MIO comes from;

Most important: the authors have shown some important supplementary data in the Response to Reviewers file only: those data should be visible to all the readers, not only to Referees. It should be necessary to set up a Supplementary Information file to collect all those data.
